# Dysregulated Immunological Functionome and Dysfunctional Metabolic Pathway Recognized for the Pathogenesis of Borderline Ovarian Tumors by Integrative Polygenic Analytics

**DOI:** 10.3390/ijms22084105

**Published:** 2021-04-15

**Authors:** Chia-Ming Chang, Yao-Feng Li, Hsin-Chung Lin, Kai-Hsi Lu, Tzu-Wei Lin, Li-Chun Liu, Kuo-Min Su, Cheng-Chang Chang

**Affiliations:** 1School of Medicine, National Yang-Ming University, Taipei 112, Taiwan; cm_chang@vghtpe.gov.tw; 2Department of Obstetrics and Gynecology, Taipei Veterans General Hospital, Taipei 112, Taiwan; 3School of Medicine, National Yang Ming Chiao Tung University, Taipei 112, Taiwan; 4Department of Pathology, Tri-Service General Hospital, National Defense Medical Center, Taipei 114, Taiwan; liyaofeng1109@gmail.com; 5Division of Clinical Pathology, Department of Pathology, Tri-Service General Hospital, National Defense Medical Center, Taipei 114, Taiwan; Hsinchunglin@gmail.com; 6Graduate Institute of Pathology and Parasitology, National Defense Medical Center, Taipei 114, Taiwan; 7Department of Medical Research and Education, Cheng-Hsin General Hospital, Taipei 112, Taiwan; lionel.lu@gmail.com; 8Department of Medical Research, Taipei Veterans General Hospital, Taipei 112, Taiwan; backyard0826@gmail.com; 9Department of Obstetrics and Gynecology, Tri-Service General Hospital, National Defense Medical Center, Taipei 114, Taiwan; lvita.tw@gmail.com; 10Division of Obstetrics and Gynecology, Tri-Service General Hospital Songshan Branch, National Defense Medical Center, Taipei 105, Taiwan; 11Graduate Institute of Medical Sciences, National Defense Medical Center, Taipei 114, Taiwan

**Keywords:** borderline ovarian tumors (BOTs), gene ontology (GO), functionome-based and data-driven analysis, immune and inflammatory response, cell membrane and transporter, cell cycle and signaling, cell metabolism, galactose catabolism

## Abstract

The pathogenesis and molecular mechanisms of ovarian low malignant potential (LMP) tumors or borderline ovarian tumors (BOTs) have not been fully elucidated to date. Surgery remains the cornerstone of treatment for this disease, and diagnosis is mainly made by histopathology to date. However, there is no integrated analysis investigating the tumorigenesis of BOTs with open experimental data. Therefore, we first utilized a functionome-based speculative model from the aggregated obtainable datasets to explore the expression profiling data among all BOTs and two major subtypes of BOTs, serous BOTs (SBOTs) and mucinous BOTs (MBOTs), by analyzing the functional regularity patterns and clustering the separate gene sets. We next prospected and assembled the association between these targeted biomolecular functions and their related genes. Our research found that BOTs can be accurately recognized by gene expression profiles by means of integrative polygenic analytics among all BOTs, SBOTs, and MBOTs; the results exhibited the top 41 common dysregulated biomolecular functions, which were sorted into four major categories: immune and inflammatory response-related functions, cell membrane- and transporter-related functions, cell cycle- and signaling-related functions, and cell metabolism-related functions, which were the key elements involved in its pathogenesis. In contrast to previous research, we identified 19 representative genes from the above classified categories (IL6, CCR2 for immune and inflammatory response-related functions; IFNG, ATP1B1, GAS6, and PSEN1 for cell membrane- and transporter-related functions; CTNNB1, GATA3, and IL1B for cell cycle- and signaling-related functions; and AKT1, SIRT1, IL4, PDGFB, MAPK3, SRC, TWIST1, TGFB1, ADIPOQ, and PPARGC1A for cell metabolism-related functions) that were relevant in the cause and development of BOTs. We also noticed that a dysfunctional pathway of galactose catabolism had taken place among all BOTs, SBOTs, and MBOTs from the analyzed gene set databases of canonical pathways. With the help of immunostaining, we verified significantly higher performance of interleukin 6 (IL6) and galactose-1-phosphate uridylyltransferase (GALT) among BOTs than the controls. In conclusion, a bioinformatic platform of gene-set integrative molecular functionomes and biophysiological pathways was constructed in this study to interpret the complicated pathogenic pathways of BOTs, and these important findings demonstrated the dysregulated immunological functionome and dysfunctional metabolic pathway as potential roles during the tumorigenesis of BOTs and may be helpful for the diagnosis and therapy of BOTs in the future.

## 1. Introduction

Ovarian low malignancy potential (LMP) tumors, or borderline ovarian tumors (BOTs), are a unique subtype of epithelial ovarian cancers (EOCs) that are the leading cause of death of gynecologic cancers and the fifth leading cause of all cancer-related deaths among women. To date, BOTs consist of disparate groups of neoplasms based on histopathological and molecular characteristics, as well as clinical behavior [1]. In 1929, Howard Taylor first found ovarian LMP tumors as a “semimalignant” disease between benign neoplasms and invasive carcinoma, regardless of clinical manifestations or histologic features [2]. In 1971, the International Federation of Obstetrics and Gynecology (FIGO) identified these “semimalignant” ovarian tumors as a “low-grade malignant tumor” totally different from ovarian cancer, and the word “borderline tumor” displaced “low-grade malignant tumor” in the World Health Organization (WHO) classification of female genital tumors in 2014 [3,4]. The BOT accounts for approximately 10~15% of EOCs and usually occurs in younger women compared with generally common high-grade serous ovarian, tubal, and peritoneal cancers with a stepwise manner of progression from precursor lesions to invasive disease [5,6]. In recent decades, clear evidence has shown that the BOT does have intercalary biologic, histologic, pathogenetic, and molecular features intermediate between clearly benign and frankly malignant ovarian neoplasms, and BOTs are classified into serous borderline ovarian tumors (SBOTs), mucinous borderline ovarian tumors (MBOTs), seromucinous borderline tumors, endometrial borderline tumors, clear cell borderline tumors, transitional (Brenner) and other subtypes on the basis of histogenesis and histopathology in light of the recent 2014 WHO classification of tumors of female genital organs [4]. Generally, SBOTs, approximately 65% of BOTs [7], occur mostly in North America, the Middle East, and most of Europe. In contrast, MBOTs, approximately 32% of BOTs [8], occur predominately in East Asia and parts of Europe [9]. BOTs are predominantly diagnosed in premenopausal females before the age of 40 and are rarely confirmed in older women after the age of 65 [10,11,12].

BOTs, unlike invasive ovarian cancer, are chiefly diagnosed at an early stage (75% of BOTs are diagnosed at stage I) and have more slothful clinical behavior, resulting in an excellent prognosis [13]. Generally, the 5-year overall survival rates of BOTs for stages I, II, and III are 99, 98, and 96%, respectively, and the 10-year overall survival rates of BOTs for stages I, II, and III are 97, 90, and 88%, respectively [14]. Nevertheless, some patients with BOTs under primary treatment may suffer from later symptomatic recurrence or malignant transformation resistant to platinum chemotherapy and death even after 20 years [15,16,17]. To date, surgery is still the major ideal method to treat BOTs. There are two standard surgical methods applied for removing macroscopically visible BOTs: a conservative operation for the young with a desire for fertility preservation or childbearing and radical surgery for the postmenopausal groups [18,19,20]. In addition, the method of surgery also depends on the histopathological characteristics of BOTs since there is always a risk of recurrence or development of invasive ovarian tumors [13,21,22]. Adjuvant chemotherapy and radiotherapy are not usually considered the standard therapy except surgical intervention because the role of adjuvant chemotherapy for BOTs is restricted [23,24,25,26,27], not to mention contentious radiational, hormonal, or targeted therapy in borderline ovarian tumors [21].

Because conventional chemotherapy demonstrated very limited activity in BOTs, some studies had started on molecular and gene mutations of BOTs early, hoping to find a more effective treatment for eradication after surgery. Recent studies have inferenced several assumptions, including the incessant ovulation hypothesis, gonadotropin hypothesis, hormonal hypothesis, and inflammation hypothesis, for the tumorigenesis of BOTs [28,29,30,31]. Earlier studies for BOTs and their first two most common subtypes, SBOT and MBOT, identified that mutations in the *KRAS*, *BRAF*, and *ERBB2* genes and overexpression of the *p53* and *Claudin-1* genes characterized SBOTs and that *KRAS* mutation, *ERBB2* mutation or amplification, *trefoil factor-3 (TFF3)* strong expression, and *HER-2/neu* amplification accounted for a certain proportion of MBOT occurrence [17,32,33,34,35,36,37]. Therefore, detecting the status of *KRAS*, *ERBB2*, *p53*, or *BRAF* mutation status may be a useful way to predict or investigate the possibility and tendency for the recurrence of BOTs or invasive ovarian carcinoma under satisfactory clinical scenarios. In addition to discovering the position of gene mutations, several preclinical studies have also clarified that several biomolecular activations of the mitogen-activated protein kinase (MAPK)/extracellular signal-regulated kinase (ERK) pathway, PI3K/AKT/mTOR pathway, Hedgehog pathway, and angiogenesis pathway take place in both SBOTs and MBOTs, with certain separate proportions that may appear as targeting subjects for innovative therapies [17,34,38,39,40,41]. As a result, clinical trials for low-grade serous carcinoma, mucinous carcinoma, and BOTs with targeted therapies focused on potential genes or pathways mentioned above have developed slowly and gradually in recent years, including MEK inhibitor (MEKi) therapy, agents targeting the PI3K/AKT/mTOR pathway, and antiangiogenic agents. However, all results of these targeted therapies are still pending to date due to extremely limited information and experience [42,43,44].

Although modern molecular studies, including analyses of mutational status, DNA copy number changes, and gene expression profiles, have provided initial insight into the pathogenesis of BOTs, there is still no integrative model for analyzing the comparison of genome profiles between BOTs, the more common SBOT, and the less common MBOT. To further understand the detailed information of the crucial deregulated biomolecular and genetic functions, we had previously organized integrated gene expression profiles downloaded from public databases and established a gene-based set regularity model based on the ordinal change among the gene elements in a gene set detected by microarrays to rebuild the gene set regularity (GSR) indices of the global functions and functionomes for measurement of the changes of the ordering levels of the gene elements defined by the gene ontology (GO) gene sets definition. It could be utilized for investigating the meaningful dysregulated functions and dysfunctional pathway participating in complicated diseases such as ovarian carcinomas via comparison to differentially expressed genes (DEGs) [45,46,47,48,49,50]. Using these research methods, we conducted a genome-wide integrative investigation to analyze the global functions of BOTs at different subtypes by analyzing entire meaningful deregulated functions of BOTs detected by microarrays via concepts of the DEGs and rebuilt a functionome-pattern of a GSR model of the global functions to investigate further comprehensive data of the related, meaningful, neoplastic mechanisms and dysregulated functions accompanied by corresponding genes at different subtypes of BOTs, including mainly SBOTs and MBOTs. We could observe more clearly if there was significant functional pathogenic and biomolecular deterioration among all BOTs, SBOTs, and MBOTs by quantifying the general and further categorized functions under the structure of the GO-defined gene sets. The consequences of these analytics may be conducive to further specific investigation and advancement of precise therapy for BOTs in the future.

## 2. Results

### 2.1. Workflow for the Integrative Analyzing Model

The workflow of this study is shown in Figure 1, and the minutiae of this algorithm are depicted in the Section 4. First, we transformed the extracted gene expression profiles of the gene elements to ordinal data and then to 10,192 quantified GO terms based on the sequence of expression from the gene elements in each gene set. This procedure generated functionomes consisting of 10,192 GSR indices, defining relatively comprehensive biological and molecular functions for investigating BOTs. Next, we calculated the quantified functions and the functional regularity patterns between 92 BOT samples and 136 normal ovarian controls with gene set regularity (GSR) indices and set up the GSR model of the functionome pattern. We then investigated the informativeness of the genome-wide functionome consisting of GSR indices and established a functionome-based training method for classification and prediction with the help of a set of supervised mathematical commands from machine learning, which is a support vector machine (SVM). The deregulated functions were detected by significant difference between BOT groups and normal controls, and the *p*-value was set at 0.05. The variation in the GSR indices between each BOT group consisting of all BOTs, SBOTs, and MBPTs and normal controls indicated that the biomolecular functions were widely dysregulated in the BOT groups compared with the normal controls with statistical significance. Finally, we performed a whole-genome integrative analysis to identify meaningful dysfunctional pathways and potential DEGs and possible essential parts of the pathogenesis of BOTs by discovering the dysregulated biomolecular and genetic functions of BOTs detected by microarrays with gene expression profiles. The crucial biological functions and genes involved in the pathogenesis of BOTs were detected by investigating genome-wide GO-defined functions and DEGs.

### 2.2. Microarray Gene Expression Datasets and Gene Set Definition

We used the integrative method of gene ontology-based analysis to investigate all related dysregulated functions of BOTs. DNA microarray gene expression datasets were downloaded from the National Center for Biotechnology Information (NCBI) Gene Expression Omnibus (GEO) database, and the sample data were obtained from 28 dataset series containing six different DNA microarray platforms without any missing data. A total of 92 BOT samples were collected, including 79 samples of SBOT and 13 samples of MBOT in terms of histological classification (Table 1); 35 samples of stage I, five samples of stage II, ten samples of stage III, one sample of stages IV, and 41 samples of unconfirmed stages in terms of the International Federation of Gynecology and Obstetrics (FIGO) staging system. For comparison, 136 normal ovarian samples were gathered as a control group. Detailed information of all collected samples is available in Appendix A. The 10,192 GO gene set definitions for annotation of all functions were downloaded from the Molecular Signatures Database (MSigDB) with the version “c5.all.v7.1.symbols.gmt” [51].

### 2.3. Means and Histograms of GSR Indices of Functionomes among Each BOT Sample Group with Different Divergences

We calculated the means of the GSR indices of functionomes for all BOTs, serous BOTs (SBOTs), and mucinous BOTs (MBOTs), corrected by the averages from the control groups, as shown in Figure 2. The GSR index delineated by quantified changes in the gene expression ranking in a gene set was computed based on the extent of ranking change within a gene set defined by the GO terms or biological canonical pathways between the case and control groups, and the variations in GSR indices between each case and the normal control group were statistically significant (*p* < 0.05), which revealed notably decreased deviations among all BOT groups (orange lattice in Figure 2A), SBOTs (orange lattice in Figure 2B), and MBOTs (orange lattice in Figure 2C), indicating the steady deterioration of functional regulation of BOTs apart from normal controls (blue lattice in Figure 2). More obvious deviation in MBOTs distinct from SBOTs indicated more irregular changes of dysregulated function. Quantifying the regulation of dysregulated functions by surveying the average of the total GSR indices among each functionome, with subsequent corrections based on the control groups, the numerical average values of the corrected GSR indices for all BOTs, SBOTs, and MBOTs were 0.6689, 0.7036, and 0.5032, respectively.

### 2.4. Global Functional Regularity Patterns Predicted and Classified by Machine Learning with High Accuracies

As the histograms show above, there were indeed significant differences in functional regularity patterns among the three case groups (all BOTs, SBOTs, and MBOTs) from the normal control group. With the help of supervised machine learning, we used a support vector machine (SVM) technological algorithm to recognize, classify, and predict distinct functionomes with GSR indices. The performance was evaluated with binary classification and inspected by fivefold cross-validation. The cumulative performance results are listed in Table 2 with the averages of ten continuous classifications and predictions. The accuracies of binary classification (case vs. control) ranged from 99.77% to 100.00%. The classification between the MBOT and normal control groups had the best results. The areas under the curve (AUCs) of the test for each case group ranged from 0.9967 to 1.0000. These results with high accuracies indicated that the functional regularity patterns quantified by the GSR indices converted from the microarray gene expression profiles can provide adequate informative data for the SVM to perform correct recognition and classification. The results also revealed that the functional regularity patterns among all BOTs, SBOTs, and MBOTs were distinct and could be utilized for the molecular classification of the gene expression profiles among each case group.

### 2.5. The Most Deregulated and Common Gene Ontology (GO) Terms of BOTs and Subtypes

There were 717, 655, and 792 GO terms among all BOT, SBOT, and MOBT groups, respectively, summarized and ranked by cluster weight index (CWI), which was a measured index of a weighed ratio based on the *p*-values for each clustered deregulated GO term with statistical significance. CWI was defined as the ratio of that cluster weight divided by the sum weight of total clusters to measure the weight and to represent the relevance of each cluster in the GO tree, and we used CWI calculated to quantify and evaluate the importance of each GO cluster in the pathogenesis of BOTs. A higher CWI indicates higher relevance and importance. The 50 most deregulated GO terms ranked by CWI for all BOTs, SBOTs, and MBOTs are displayed in Table 3. The first deregulated GO term for each stage group was “regulation of immune system process (GO:0002682)” for all BOTs, “regulation of immune system process (GO:0002682)” for SBOTs, and “small molecule metabolic process (GO:0044281)” for MBOTs. Then, we summarized and rearranged the top 41 GO terms in order that appeared repeatedly in the 50 most deregulated GO terms among the three groups according to the weighted CWI proportions in each group, as shown in Table 4 with their original ranking in each group. We then traced and compared the trends of ranking order of the top 41 common deregulated GO terms for each group to evaluate the importance of a given functionome among all BOTs, SBOTs, and MBOTs. The ranking trend distributions of deregulated GO terms for all BOTs, SBOTs, and MBOTs are shown in Figure 3 and the whole deregulated GO terms of all BOTs, SBOTs, and MBOTs are listed in detail in Appendix A. We found obviously that the ranking orders and trends of these top 41 common deregulated GO terms of all BOTs and SBOTs were the same in the majority, and the ranking trend distribution of the deregulated GO terms of MBOTs did not go congruously with all BOTs and SBOTs. Next, all top 41 common deregulated GO terms among the three case groups could be classified based on the functions they represent and organized into the following categories: immune and inflammatory response-related functions, cell membrane and transporter related functions, cell cycle and signaling related functions, cell metabolism related functions, and others that are not classified into the above four categories.

### 2.6. Immune and Inflammatory Response-Related Functions and Relevant DEGs

There were three GO terms among all BOT, SBOT, and MBOT groups classified into immune and inflammatory response-related functions: “regulation of immune system process (GO:0002682)”, “immune system development (GO:0002520)”, and “immune effector process (GO:0002252)”; the rankings of these immune-related GO terms in all groups were very much in front order. Each GO term, whether its representative cellular component, molecular function, or biological process, has a corresponding genome that annotates genes and gene products to enable functional interpretation of experimental data [52]. We utilized statistical methods to discover the potential genes from the genes with the definition of GO gene sets (http://geneontology.org/, accessed on 15 April 2021) annotated for the above three GO terms and then intersected the DEGs calculated separately to find the relevant DEGs with the highest repetition. We found that the two DEGs: *IL6* (interleukin 6) and *CCR2* (C-C motif chemokine receptor 2) had the most frequent occurrence. The trends of ranking orders for each categorized GO term are shown in Figure 4 with detailed information.

### 2.7. Cell Membrane- and Transporter-Related Functions and Relevant DEGs

Similarly, based on the annotated biological functions, nine GO terms among three disease groups were categorized as cell membrane- and transporter response-related functions: “ion transport (GO:0006811)”, “transporter activity (GO:0005215)”, “regulation of transport (GO:0051049)”, “locomotion (GO:0040011)”, “secretion (GO:0046903)”, “biological adhesion (GO:0022610)”, “transmembrane transport (GO:0055085)”, “cellular macromolecule localization (GO:0070727)”, and “intracellular transport (GO:0046907)”; the ranking orders and trends of these clustered GO terms are revealed in Figure 5 in detail. We also used the calculation method to identify the potential genes from the previously mentioned available database, and four relevant DEGs were identified with the highest repetition: *IFNG* (interferon gamma), *ATP1B1* (sodium/potassium-transporting ATPase subunit beta-1), *GAS6* (growth arrest specific 6), and *PSEN1* (presenilin 1).

### 2.8. Cell Cycle- and Signaling-Related Functions and Relevant DEGs

We categorized 16 of the top 41 common deregulated GO terms into cell cycle- and signaling response-related functions according to their assigned functions, and the 16 GO terms were “cell activation (GO:0001775)”, “regulation of cell differentiation (GO:0045595)”, “cytokine production (GO:0001816)”, “homeostatic process (GO:0042592)”, “positive regulation of multicellular organismal process (GO:0051240)”, “cell-cell signaling (GO:0007267)”, “negative regulation of response to stimulus (GO:0048585)”, “chromosome organization (GO:0051276)”, “apoptotic process (GO:0006915)”, “cell cycle (GO:0007049)”, “regulation of organelle organization (GO:0033043)”, “positive regulation of signaling (GO:0023056)”, “negative regulation of multicellular organismal process (GO:0051241)”, “signaling receptor binding (GO:0005102)”, “regulation of response to stress (GO:0080134)”, and “response to endogenous stimulus (GO:0009719).” Using the same calculation method as above, we selected three relevant DEGs with the most statistically frequent occurrence among all genes annotating the 16 classified GO terms: *CTNNB1* (catenin beta 1), *GATA3*, and *IL1B* (interleukin 1 beta). The results described in this section with ranking orders for each GO term among the three disease groups are presented fully in Figure 6.

### 2.9. Cell Metabolism Related Functions and Relevant DEGs

Based on the same principle, nine deregulated GO terms among three disease groups were classified into cell metabolism response-related functions: “small molecule metabolic process (GO:0044281)”, “lipid metabolic process (GO:0006629)”, “regulation of protein modification process (GO:0031399)”, “RNA metabolic process (GO:0016070)”, “regulation of phosphorus metabolic process (GO:0051174)”, “regulation of nucleobase-containing compound metabolic process (GO:0019219)”, “DNA metabolic process (GO:0006259)”, “protein phosphorylation (GO:0006468)”, and “peptidyl-amino acid modification (GO:0018193).” Likewise, by utilizing statistics and calculations, ten relevant DEGs with the highest repetition in this group were selected and noted as follows: *AKT1*, *SIRT1* (sirtuin 1), *IL4* (interleukin 4), *PDGFB* (platelet-derived growth factor subunit B), *MAPK3* (mitogen-activated protein kinase 3), *SRC*, *TWIST1* (Twist-related protein 1), *TGFB1* (transforming growth factor beta 1), *ADIPOQ* (adiponectin), and *PPARGC1A* (peroxisome proliferator-activated receptor gamma coactivator 1-alpha). All findings mentioned in this paragraph and the ranking orders with trends of the nine GO terms for each disease group are completely disclosed in Figure 7.

### 2.10. The Most Significantly Dysfunctional Canonical Pathways and Representative DEGs

We found 4833, 4065, and 5305 canonical pathways among all BOT, SBOT, and MBOT groups and arranged them in the order of relevance according to their *p*-values to discover the most dysfunctional pathway in each group. In Table 5, the top 20 most dysfunctional canonical pathways ranked by *p*-value are listed, and all *p*-values were statistically significant. “REACTOME galactose catabolism” ranked first in all BOT and SBOT groups, and “REACTOME glucuronidation” ranked first in the MBOT group. In addition, the top 20 pathways in the three disease groups had only one common dysfunctional pathway, “REACTOME galactose catabolism”, which was also ranked fourth in the MBOT group. In addition, according to the gene set data downloaded from the GESA (Gene Set Enrichment Analysis) website (https://www.gsea-msigdb.org/gsea/msigdb/cards/REACTOME_GALACTOSE_CATABOLISM.html, accessed on 15 April 2021), we found five genes related to this dysfunctional pathway and their corresponding proteins (*GALE*, *GALK1*, *GALT*, *PGM2*, and *PGM2 L1*). Next, we used functional protein association networks (https://string-db.org/, accessed on 15 April 2021) to compare these five proteins for the correlation of interactions with each other and found that *GALT* (galactose-1-phosphate uridylyltransferase) was the most promising gene and protein.

### 2.11. Verification with Immunohistochemical Analysis of Anti-IL6 and Anti-GALT Expression between BOTs and Normal Ovarian Tissues

To verify the abovementioned related pathogenic mechanisms and to explore clinical manifestations specifically of the identified DEGs involved in the tumorigenesis process of BOTs, we collected related clinical samples from a cohort (BOTs, *n* = 9; normal control group, *n* = 9) and then performed immunostaining with anti-IL6 and anti-GALT antibodies separately. Among all deregulated GO terms and classified categories, the general ranking and order occupied by the immune and inflammatory response-related functions was the highest overall; as a result, we picked up the IL6 gene selected from the immune and inflammatory response-related functions for the representative gene among 19 DEGs due to its close relationship with ovarian tumors [53,54,55,56]. Together with the anti-GALT antibody, we used immunohistochemical analysis of the anti-IL6 antibody between the BOT and control groups to assess the clinically meaningful significance of IL6 and GALT. During the entire process, the pathologists interpreted and verified to obtain the following results on average repeatedly. The results revealed higher expression levels of IL6 and GALT in BOT samples than in normal samples (Figure 8A). Pathologists and quantification of the immunostaining scores of IL6 and GALT levels were performed using SPSS software (IBM Corp., Released 2013, IBM SPSS Statistics for Windows, Version 22.0, Armonk, NY, USA: IBM Corp.). A higher mean value with statistical significance of IL6 and GALT expression in samples of BOTs was clearly shown compared with the control group (Figure 8B). These results supported the above deductions, indicating that many deregulated functions inferred from integrative GO enrichment analysis, including immune and inflammatory response-related, cell membrane-related and transporter-related, cell cycle- and signaling-related, and cell metabolism-related functions, indeed contributed to the pathogenesis of BOTs. Likewise, these results also proved that the dysfunctional pathway of galactose catabolism played a role in the tumorigenesis of BOTs. All results provide preliminary clinical evidence to support the previously proposed pathogenic tumorigenesis of BOTs.

## 3. Discussion

In this study, we utilized an integrated polygenic analytical model with the concept of GO (gene ontology) and GSR (gene set regularity) indices computed based on gene expression ranking to further explore the complex and diverse biomolecular and genetic functions of BOTs and their two most common subtypes (SBOT and MBOT) and obtained statistically meaningful results. Although both subtypes belong to BOTs, serous and mucinous are different in clinical features, histopathology, and pathogenesis, although there are many commonalities found in this research. For example, in serous BOTs, clinically, there are sometimes multiple intra-abdominal metastasis-like lesions similar to ovarian cancer. In contrast to the serous subtype, mucinous BOTs usually form a single large tumor shape. In recent years, in tumorigenesis research, the epithelial-to-mesenchymal transition (EMT) has been a key concept in research on ovarian neoplasms and is a reversible process in cell differentiation, morphogenesis, growth, and change of function in which epithelial cells obtain mesenchymal cell characteristics by losing polarity and adhesion of cells and increasing cellular migratory motility [57,58]. Although BOTs are not entirely recognized as malignant tumors and are not seen as a necessary process of transformation into ovarian cancer, in this study, we found some biomarkers related to EMTs, such as IL6, AKT1, MAPK3, SRC, TWIST1, and TGFB1 [59,60,61]. These additional findings may be worthy of further investigation in the future [62]. Next, we discuss and explain the major categories mentioned in this research.

Regarding immunity- and inflammation-related functions, whether modulating or regulating the immune system process (GO:0002682), progressing to the development of the immune system against internal and invasive threats (GO:0002520) or any process of the immune system potentially contributing to immune responses (GO:0002252), the rankings of GO terms among the three disease groups were very high, and we also found that the most significant DEGs selected by repeated comparisons were IL6 and CCR2. In addition, the cell cycle, an important biological periodic function, is involved in the process of tumorigenesis, and it is worth studying further. In addition, message transmission, whether biogenetic signal transduction occurs inside the cell, across the cell membranes, or between cells, also takes a specific place in the pathogenesis of BOTs. In this study, we sorted out that transportive movement of ions (GO:0006811), active ability of transporters (GO:0005215), modulation of movement of substances (GO:0051049), self-propelled movement of a cell or organism (GO:0040011), controlled biological emancipation (GO:0046903), attachment of a cell or organism to another cell or other organism (GO:0022610), transport of a solute across a lipid bilayer (GO:0055085), macromolecule transported forward to any specific location of a cell (GO:0070727), and the directed movement of substances within a cell (GO:0046907) as cell membrane- and transporter-related functions and assembled four associated relevant DEGs (*IFNG*, *ATP1B1*, *GAS6*, *PSEN1*). In addition, we also sorted out 16 clearly defined GO terms (“Cell activation (GO:0001775)”, “Regulation of cell differentiation (GO:0045595)”, “Cytokine production (GO:0001816)”, “Homeostatic process (GO:0042592)”, “Positive regulation of multicellular organismal process (GO:0051240)”, “Cell-cell signaling (GO:0007267)”, “Negative regulation of response to stimulus (GO:0048585)”, “Chromosome organization (GO:0051276)”, “Apoptotic process (GO:0006915)”, “Cell cycle (GO:0007049)”, “Regulation of organelle organization (GO:0033043)”, “Positive regulation of signaling (GO:0023056)”, “Negative regulation of multicellular organismal process (GO:0051241)”, “Signaling receptor binding (GO:0005102)”, “Regulation of response to stress (GO:0080134)”, and “Response to endogenous stimulus (GO:0009719)”) as mentioned earlier that were classified as cell cycle- and signaling-related functions and used statistical methods to find three highly related DEGs (*CTNNB1*, *GATA3*, and *IL1B*). We also confirmed the role of cell metabolism in the pathogenetic process of BOTs. Deregulated genetic expression and dysfunctional reprogramming pathways of cell metabolism accompanied by both direct and indirect consequences of tumorigenic mutations cause tumorigenesis. Gene expression, cellular differentiation, and the tumor microenvironment are all influenced by associated metabolic reprogramming and intracellular and extracellular metabolites to acquire necessary nutrients [63]. In this study, we sorted out that cellular metabolic processes involving deoxyribonucleic acid (GO:0006259), ribonucleic acid (GO:0016070), regulation of nucleobase-containing compounds (GO:0019219), lipids (GO:0006629), and small molecules (GO:0044281), as well as alteration and modification of peptidyl-amino acid (GO:0018193), protein phosphorylation (GO:0006468), regulation of protein modification (GO:0031399), and phosphorus metabolic process (GO:0051174) are very meaningful and the selected ten DEGs do have roles in cell metabolism and tumorigenesis of BOTs.

This article also analyzed and investigated related DEGs, and the following is a brief description of each DEG. In addition to utilizing an immunohistochemical method to verify that IL6, a potential regulatory biomarker, in BOTs does have a significantly higher performance than controls, IL6 has certain accessorial functions associated with nearby immune-related cells in the microenvironment for the enhancement of benign or malignant ovarian tumors [53,54,55,56,64,65]. CCR2 can bind with chemokine ligand 2 (CCL2) on macrophages and monocytes to promote the expression and differentiation of T cells and to regulate related inflammatory cytokines by activating the PI3K cascade and small G protein GTPases. CCR2 plays a certain role in the tumorigenesis of ovarian tumors through immune cells and related regulatory responses in the microenvironment, which is also worthwhile to explore in the future [17,56,66,67,68,69]. IFNG (IFNγ), the only type II interferon, is an important signal activator of macrophages and inducer of innate and adaptive immunity in accordance with other immune-related cells or cytokines to exert antiproliferative and antitumor effects [66,70,71]. ATP1B1 is mainly responsible for establishing and maintaining the electrochemical gradients of Na and K ions across the plasma membrane, and the expression of *ATP1B1* may have a suppressive effect that needs further validation [72,73]. GAS6 is usually thought to be related to cell proliferation, chemotaxis and survival, and stronger expression in BOTs implies associated epithelial-mesenchymal transition in the tumor microenvironment [74,75]. PSEN1 (Presenilin-1, PS-1) is considered to regulate amyloid precursor protein (APP) catalysis and is thought to be involved in Notch and *Wnt* signaling cascades; however, the role of PSEN1 in BOTs is rarely known and needs to be further validated [76,77]. CTNNB1 (β-catenin) is a crucial component of the *Wnt* signaling pathway for the coordination and regulation of gene transcription and cell–cell adhesion through phosphorylation and mutation of the *CTNNB1* gene, and alteration of the β-catenin pathway is thought to participate in the pathogenic process of BOTs [78,79,80,81]. GATA3 is considered a transcriptional activator for the regulation of T-cell development and several inflammatory and allergic responses and was suggested as a regulator in benign ovarian Brenner tumors and serous tumors with prognostic value [4,82,83]. IL1B (interleukin 1 beta, IL-1β) is a potent proinflammatory mediator of the inflammatory response, as well as having a relationship in cell proliferation, differentiation, and apoptosis, whereas IL1B can promote tumorigenesis through the inflammatory response, which still needs further verification among BOTs [84,85,86]. AKT1, also known as Akt/PKB, participates in the PTEN/PI3K/Akt signaling pathway to mediate apoptosis, metabolism, cell proliferation, and growth and is frequently dysregulated during tumorigenesis [81,87]. SIRT1 (Sirtuin1), a class III histone deacetyltransferase and a nicotinamide adenine dinucleotide-dependent deacetylase, exerts protective effects against oxidative stress, genomic instability, and DNA damage, and *SIRT1* overexpression may play a crucial role in the tumorigenesis of BOTs and the carcinogenesis of early-stage ovarian cancer [88,89,90]. IL4 (interleukin 4), as a key regulator in humoral and adaptive immunity, could induce Th2 cell differentiation and may have a significant effect on the pathogenesis of BOTs [56,91]. PDGFB (platelet-derived growth factor B) regulates cell growth and division and plays an important role in neoangiogenesis, particularly in tumorigenesis among both BOTs and EOCs [92,93,94,95]. MAPK3 (mitogen-activated protein kinase 3), also known as ERK1 (extracellular signal-regulated kinases 1), acts in the mitogenic *Ras/Raf/MEK/ERK* signaling cascades to regulate cell proliferation, differentiation, and cell cycle progression, and MAPK3 is a key factor for the pathogenesis of BOTs accompanied by a high prevalence of *KRAS* and *BRAF* mutations [33,81,96,97]. SRC, a nonreceptor protein tyrosine kinase, participates in the regulation of cell growth and embryonic development and promotes survival, angiogenesis, proliferation and invasion pathways while being activated or overexpressed during tumor development [43,98,99]. TWIST1, acting as an epithelial-mesenchymal transition transcriptional regulator, plays an important role in escaping apoptosis and metastasis, with increased expression stepwise from benign, borderline, to malignant ovarian tumors [100,101,102]. TGFB1 could conduct several cellular functions, such as cell proliferation, differentiation, growth, and apoptosis, and TGFB1 could also work with other immune-related cells in the microenvironment to show proangiogenic and prometastatic features in addition to the dual role of being an enemy and friend in tumorigenesis [56,103,104]. ADIPOQ (adiponectin), mainly produced and associated with adipose tissue, modulates numerous metabolic processes consisting of regulation of glucose level, metabolism of fatty acids, and insulin sensitivity and may take some part in cell growth, angiogenesis, and tissue remodeling during tumorigenesis with significantly lower levels in cancerous ovarian samples [105,106,107,108]. PPARGC1A (PGC-1α) participates essentially in metabolic reprogramming involved in gluconeogenesis, fatty acid metabolism, and mitochondrial biogenesis and facilitates a flexible metabolic profile to benefit tumor cells overexpressing PPARGC1A in human epithelial ovarian cancer cells [109,110,111,112]. Furthermore, we found the most common and meaningful dysfunctional canonical pathways among the three disease groups, galactose metabolism (REACTOME GALACTOSE CATABOLISM), and discovered a representative DEG, *GALT*. In the human liver, galactose is converted into glucose-6-phosphate through the biophysiological reaction galactose metabolism for rapid conversion from galactose to glucose, and GALT participates in a prominent position for catalyzing the second step of the Leloir pathway of galactose metabolism [113]. Galactosaemia caused by mutation of *GALT* or *GALT* deficiency could induce ovarian toxicity, and polymorphism of *GALT* with galactose consumption and metabolism may be associated with the development of BOTs and the risk of EOCs [114,115,116,117,118,119]. However, the roles of all DEGs and concomitant influences mentioned above in the tumorigenesis of BOTs require additional research data to explore further and to confirm definitively.

The limitations of this study are as follows. The first limitation is the distribution of case groups. The numbers of SBOTs and MBOTs are very different, and SBOTs are more common because most of the populations sampled by the database in this experiment are patients from Western countries and most of the patients with BOTs from Western countries are SBOTs; therefore, the presentative consequence would cause the results of the SBOT group to be similar to those of the All-BOTs group but very different from those of the MBOT group. Perhaps in the future, a more complete gene expression profile database could be established to reduce individual differences between ethnic groups in a prospective or retrospective cohort study on a larger scale globally or in parts of Eastern Asian countries. Second, some limitations of the integrative analysis model used in this research are observed including that not all human functions were concluded or defined from the gene set databases of the GO terms and the canonical pathway. The probable detectability of the GSR model was noted owing to unchanged GSR indices and missed errors during the process of converting levels to the ordering of gene expression if the expression levels were undetectable, false positivity was made from the indistinguishable elements of disparate gene sets, and the heterogenicity of different cellular histopathological compositions in tumor and control samples was utilized. Although coupled with the statistically significant high sensitivity, specificity, and accuracy of this experiment, these deficiencies may not be apparent in the performance of the overall results. As a result, more accurate programming syntax design and more specific screening of samples could be requested to avoid these problems in the future. Finally, our study mainly explored the common pathogenic mechanisms among BOTs; however, it was slightly hindered by the extracted data collected from the GO database due to insufficient research at present and the gathered clinical specimens with fewer numbers and limited funds. However, the results were statistically significant and distinctly clear in terms of clinical verification by immunostaining. More extensive BOT specimens, more worldwide academic research databases of various subtypes, and more detailed in-depth comparisons of the differences in pathogenic mechanisms of different subtypes with large-scale funding and experimental tests may be required to continue in the future.

## 4. Materials and Methods

### 4.1. Computing the GSR Indices and Reconstructing the Functionome

The GSR index was computed and extracted from the gene expression datasets by modifying the differential rank conservation (DIRAC) algorithm [120], which measures the changes of the ordering among the gene elements in a gene set between the gene expression datasets of all BOTs, SBOTs, MBOTs, and the most common gene expression ordering in the normal control samples. The details of the GSR model and the computing procedures are described in our previous studies [45,46,47,48,49,50]. Microarray gene expression datasets for BOTs and normal ovarian samples were downloaded from the GEO database. The corresponding gene expression levels were extracted and built according to the gene elements in the GO gene set and converted to ordinal data based on their expression levels. The GSR index is the ratio of gene expression ordering in a gene set between the case and the most common gene expression ordering among the normal ovarian tissue samples that ranges from 0 to 1, where 0 indicates the most dysregulated state of a function, i.e., oppositely ordered gene set regularities between BOT cases and the normal state; while 1 indicates that the regularity in a gene set remains the same between the case and the most common gene expression orderings in the normal controls. The measurement of GSR indices was carried out in the R programming language. A functionome is defined as the complete set of biological functions, and the definition for comprehensive biological functions is not yet available at present; as such, we annotated and pronounced the human functionome by using the 10,271 GO gene set defined functions. As a result, the functionome utilized in this study is defined as the assembly of 10,271 GSR indices for each sample.

### 4.2. Microarray Dataset Collection

The selection criteria for the microarray gene expression datasets from the GEO database were as follows: (1) the BOT disease samples and normal control samples should originate from ovarian tissue; (2) the datasets should provide sufficient information about the diagnosis and clinical histopathological subtypes of BOTs; and (3) any gene expression profile in a dataset was abandoned if it contained missing data.

### 4.3. Statistical Analysis

The Mann–Whitney U-test was used to test the differences among all BOTs, the major two BOT subtype groups and the controls and then corrected by multiple hypotheses using the false discovery rate (Benjamini–Hochberg procedure). The *p*-value was set at *p* < 0.05.

### 4.4. Classification and Prediction by Machine Learning

The function “kvsm” provided by the “kernlab” (version 0.9–27; Comprehensive R Archive Network; https://cran.r-project.org/, accessed on 15 April 2021), an R package for kernel-based machine-learning methods, was used to classify and predict the patterns of the GSR indices. The accuracies of the classification and predictions by SVM were measured by k-fold cross-validation. The results of ten repeated predictions were used to assess the performance of binary classification. AUC was computed using the R package “pROC” [121]. The performance of multiclass classification was assessed using the ten repeated prediction accuracies for each of the BOT and the two BOT subtype groups (all BOTs, SBOTs, and MBOTs).

### 4.5. Set Analysis

All possible logical relationship among the dysregulated gene sets of the categorized BOT subtype groups (all BOTs, SBOTs, and MBOTs) are displayed using the R package programming (version 1.6.16; Comprehensive R Archive Network; https://cran.r-project.org/, accessed on 15 April 2021).

### 4.6. Clinical Samples and Immunohistochemistry (IHC) Analysis

The clinical samples for the present study contained nine gathered BOT cases (BOTs, *n* = 9, including three serous BOTs, five mucinous BOTs and one seromucinous BOT) and nine controlled cases (normal, *n* = 9). All tissues from the cases of BOTs were collected from women who underwent surgical treatment as their therapeutic guideline. The source of all clinical normal tissues was taken from the patients after menopause or in perimenopausal status and agreed to remove the uterus and remove bilateral ovaries and fallopian tubes because of myoma or female genital organ prolapse after explaining and signing the informed consent. The patients were diagnosed and treated, and their tissues were placed in a bank at the Tri-Service General Hospital, Taipei, Taiwan. The Institutional Review Board of the Tri-Service General Hospital, National Defense Medical Center approved the study (2-107-05-043, approved on 26 October 2018; 2-108-05-091, approved on 20 May 2019). Informed consent was acquired from all patients and control subjects. All clinical tissue samples were confirmed under quantitative histopathology diagnosed by pathologists, and the assessment of the results of the immunohistochemical staining with quantitative scoring methods was scored by multiplying the intensity (I) by the percentage of positive cells (P) for all biomarkers used in this study (formula is shown as IHC score (Q) = I × P; maximum = 300) [122,123]. The detailed results of all scores for GALT and IL among clinical samples are listed in Appendix A.

## 5. Conclusions

Borderline ovarian tumor (BOT) is a peculiar subtype of epithelial tumors of the ovary with intermediate features between benign ovarian neoplasms and invasive ovarian carcinomas. In this study, we properly made use of integrative gene ontology-based analysis to investigate the most likely biomolecular and pathogenetic mechanisms among tumorigenesis of BOTs arising from normal ovarian tissues. With the assistance of machine learning, we applied a genome-wide expression profile to evaluate the various global functionomes and related pathogenic pathways of all BOTs, SBOTs, and MBOTs compared with controls and first found that there were obvious deviations in terms of the GSR index between all case samples and the control group, especially the MBOT subtype. Interestingly, we next utilized this integrated method systematically to draw 41 significantly common GO terms from the consolidated functionome that could be classified into four crucial categories consisting of three immune and inflammatory response-related functions, nine cell membrane- and transporter-related functions, 16 cell cycle- and signaling-related functions, nine cell metabolism-related functions, and four GO terms not belonging to the first four categories. Then, 19 genes corresponding to the above biological functions with high possibility were cross-checked and sorted. Moreover, we also discovered that the dysfunctional galactose catabolism pathway played a role during the formation of all BOTs, SBOTs, and MBOTs with the top few rankings. Finally, verification by using immunohistochemistry demonstrated elevated expression of IL6 and GALT in BOTs compared with normal ovarian tissue, which supported that dysregulated immunological function and dysfunctional metabolic pathways definitely participated in the tumorigenesis of BOTs. All results of this study are statistically significant, mainly because of the high sensitivity, specificity, and accuracy of the integrative polygenic analysis; and these contributions are of considerable importance in the pathogenesis of BOTs (Figure 9). Conclusively, these findings could provide a clearer direction for understanding the pathogenic mechanisms of BOTs and contribute more potential targets for treatment, monitoring, and prevention of recurrence combined with precision medicine in the future.

## Figures and Tables

**Figure 1 ijms-22-04105-f001:**
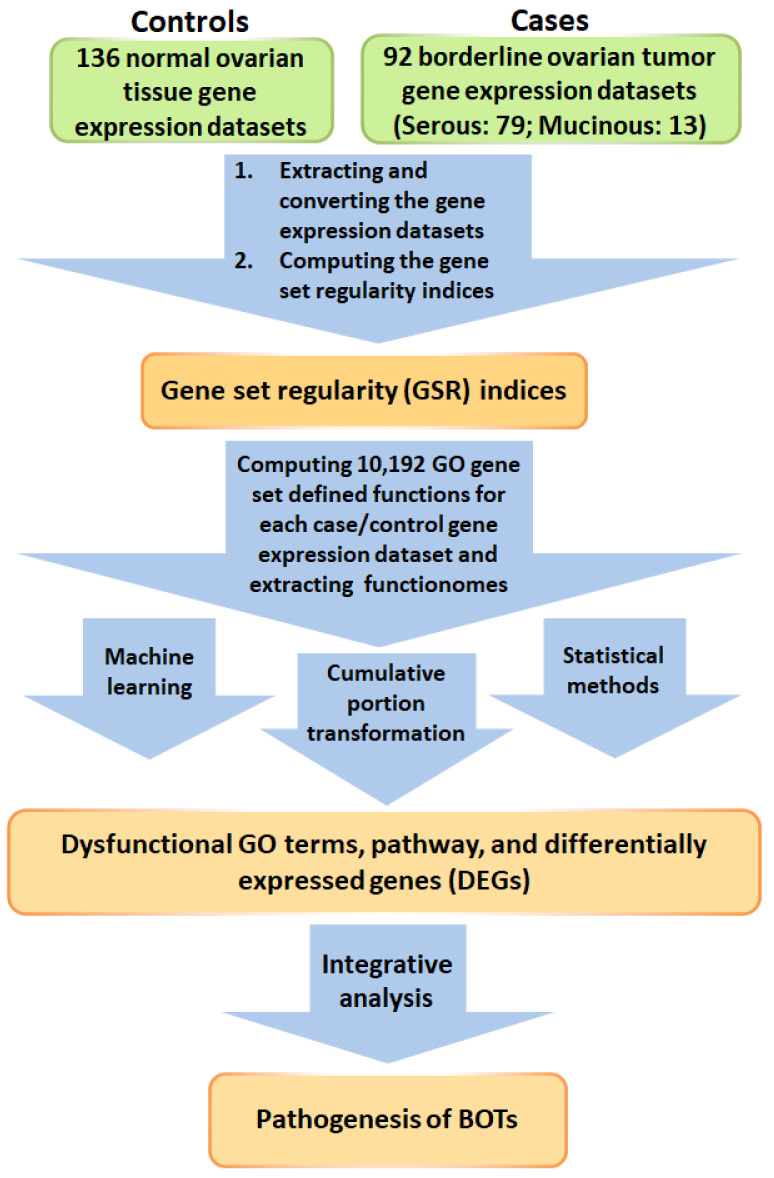
Workflow of this study. The DNA microarray gene expression datasets of 92 borderline ovarian tumor (BOT) samples and 136 normal ovarian controls were downloaded from a publicly available database with the gene set regularity (GSR) index calculated by the Gene Ontology (GO) gene set. Functionomes consisting of 10,192 GO gene sets established from the polygenic model and machine learning with statistical methods and cumulative portion transformations were used to recognize the functionome patterns to discover dysfunctional GO terms, pathways, and differentially expressed genes (DEGs). Finally, the pathogenesis of BOTs was investigated by integrative analysis.

**Figure 2 ijms-22-04105-f002:**
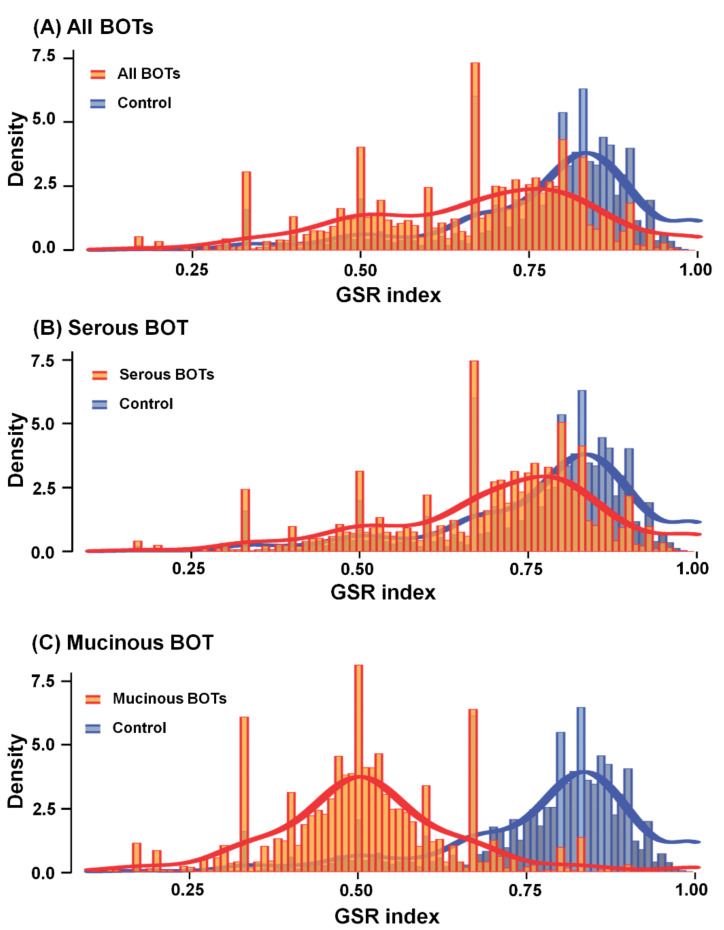
Histograms of global GSR indices of functionomes among all BOTs (serous BOTs and mucinous BOTs (orange) and control groups (blue)): These figures reveal different distributions of the functionomes for three case sample groups and control group’s statistical significance (*p* < 0.05). The normal control group (blue) located on the right side of the histogram was the same for the three case groups and used as the controls. A second peak of distribution was observed (orange), indicating deregulated biomolecular functions among all BOTs, SBOTs, and MBOTs. (**A**) Corrected GSR indices of all BOTs: 0.6689; (**B**) corrected GSR indices of serous BOTs: 0.7036; and (**C**) corrected GSR indices of mucinous BOTs: 0.5032.

**Figure 3 ijms-22-04105-f003:**
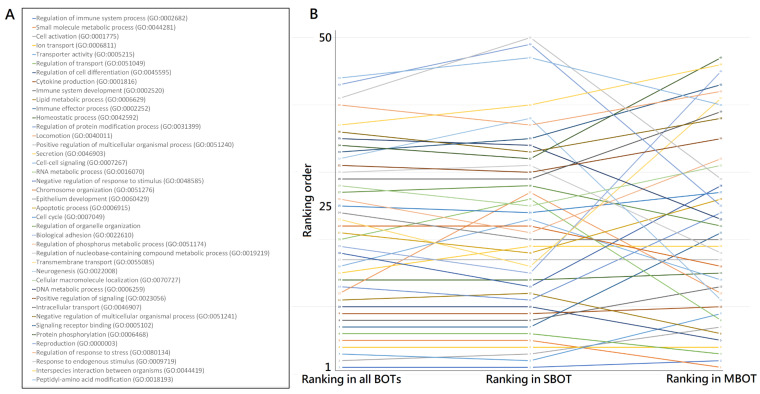
(**A**) The ranking trend distributions of deregulated GO terms for all BOTs, SBOTs, and MBOTs. The top 41 common deregulated GO terms among the three case groups are displayed in a variety of different colors, and each GO term has its corresponding ranking order. (**B**) The orders of these deregulated GO terms in SBOTs are similar to those in all BOTs, but the orders in MBOTs are very inconsistent with those of all BOTs and SBOTs.

**Figure 4 ijms-22-04105-f004:**
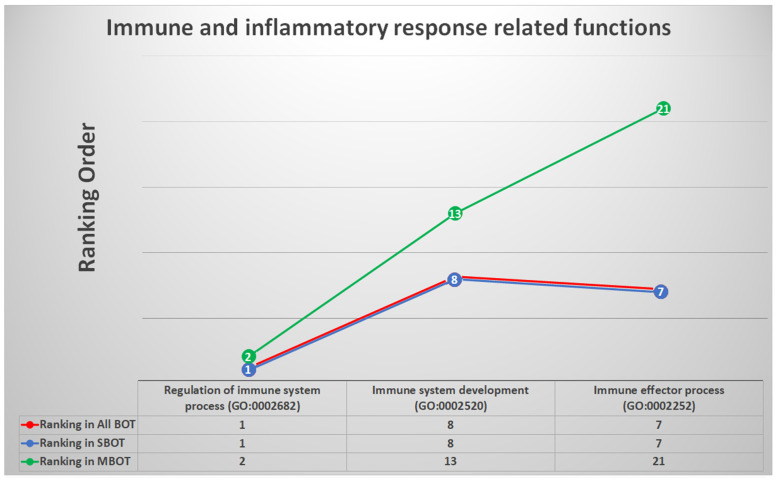
The trends and list with ranking orders of three deregulated GO terms classified into immune and inflammatory response-related functions and the two most relevant DEGs with calculations and comparison.

**Figure 5 ijms-22-04105-f005:**
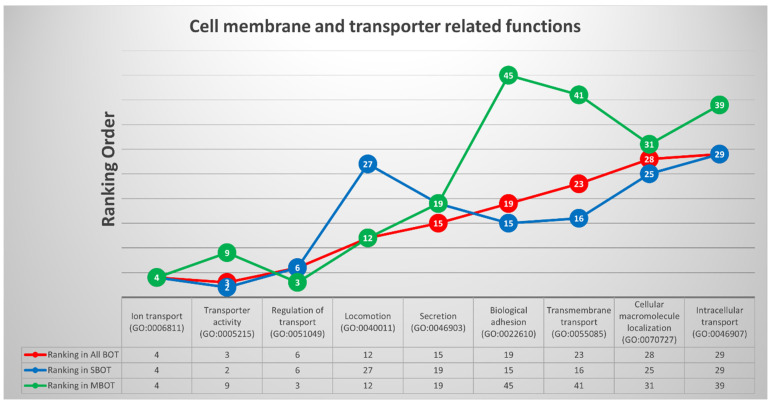
The trends and list with ranking orders of nine deregulated GO terms classified into cell membrane- and transporter response-related functions and the four most relevant DEGs with calculations and comparison.

**Figure 6 ijms-22-04105-f006:**
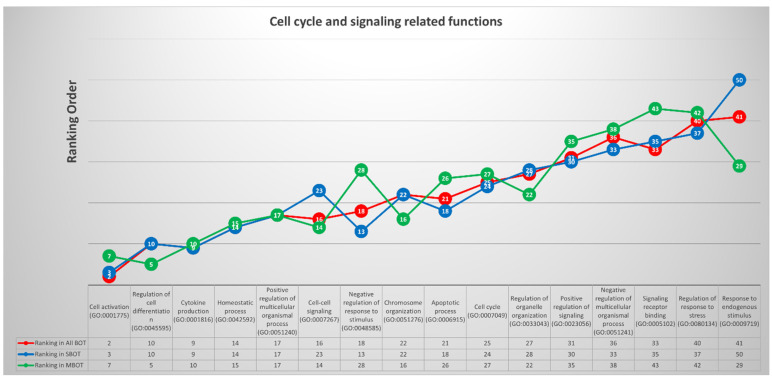
The trends and list with ranking orders of 16 deregulated GO terms classified into cell cycle- and signaling response-related functions and the three most relevant DEGs with calculations and comparison.

**Figure 7 ijms-22-04105-f007:**
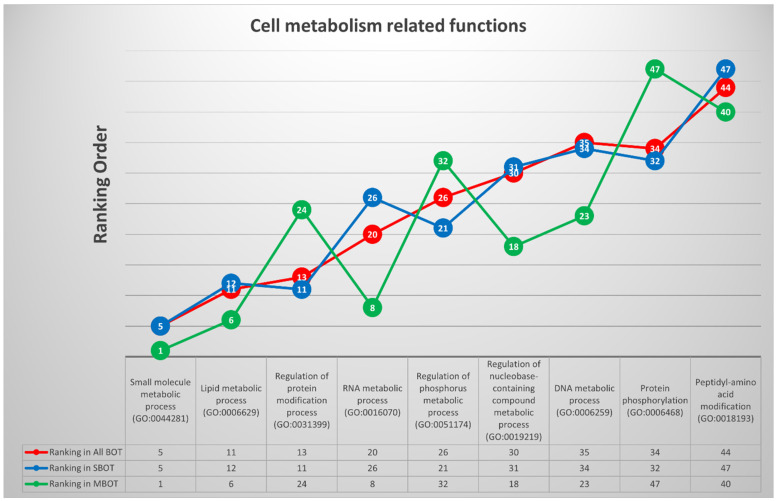
The trends and list with ranking orders of nine deregulated GO terms classified into cell metabolism response-related functions and the ten most relevant DEGs with calculations and comparison.

**Figure 8 ijms-22-04105-f008:**
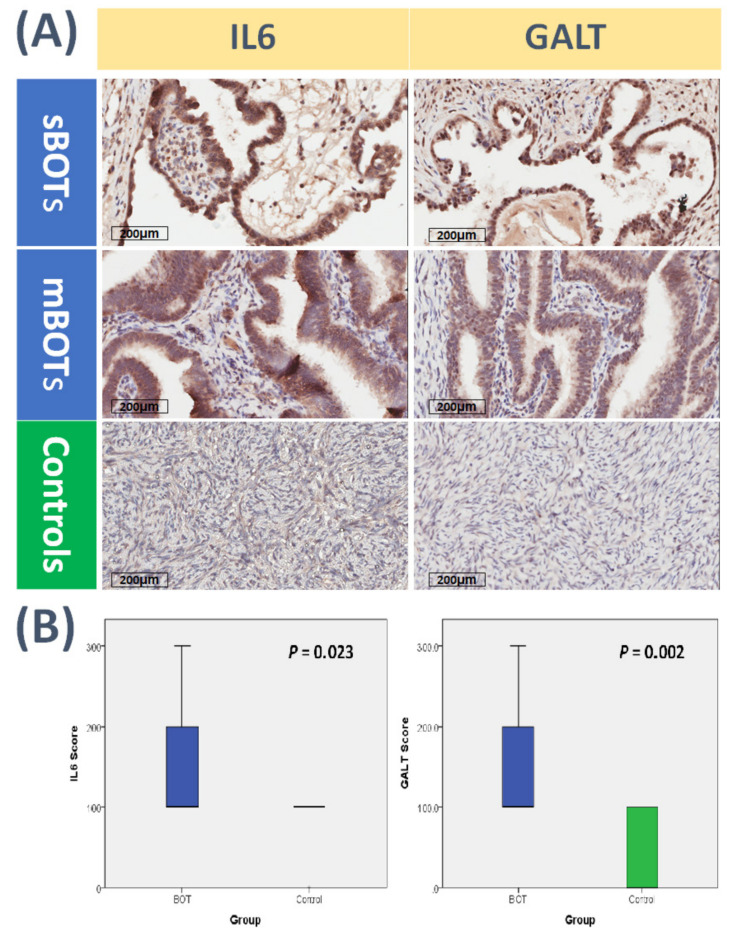
Immunohistochemical analysis of clinical samples between patients with BOTs and the control group. (**A**) Clinical samples from patients with BOTs (*n* = 9) and from the normal group (*n* = 9) were immunostained with anti-IL6 antibody (brown color, left straight row) and anti-GALT antibody (brown color, right straight row), and the parts shown in the horizontal row are the immunostaining results of serous BOTs, mucinous BOTs, and control groups from top to bottom. (**B**) Box plots for expressed IL6 and GALT between BOT patients and the control group. The expression levels of IL6 and GALT in all clinical samples are presented with quantification, and the statistically meaningful mean values of IL6 and GALT expression in the BOT group were higher than those in the normal group.

**Figure 9 ijms-22-04105-f009:**
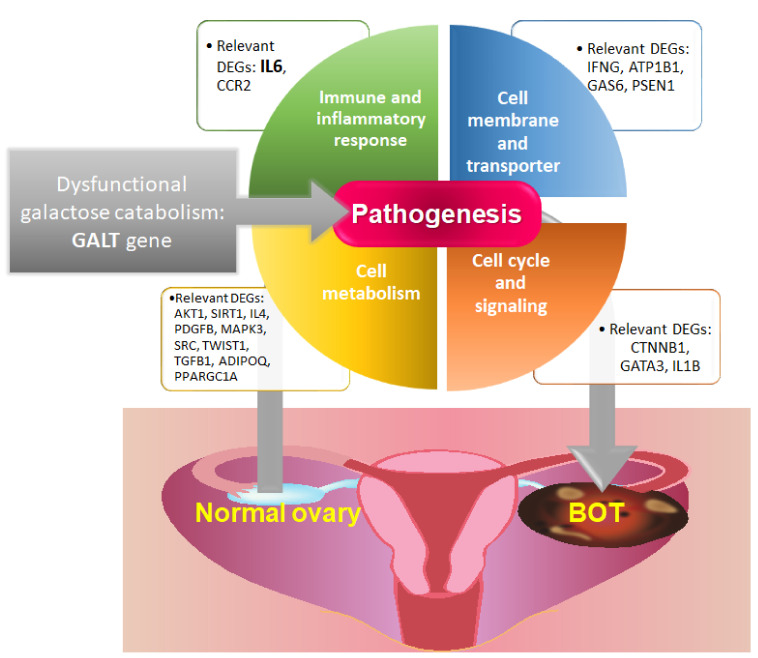
The proposed pathogenetic mechanisms involved in the pathogenesis of BOTs.

**Table 1 ijms-22-04105-t001:** Number of samples and statistics of gene set regularity indices for the BOT, serous BOT (SBOT), and mucinous BOT (MBOT) groups.

Groups	Sample	Control	Total	Sample Mean (SD ^1^)	Control Mean (SD ^1^)	*p*-Value
All BOTs	92	136	228	0.6689 (0.1892)	0.7731 (0.1647)	<0.05
SBOT (Serous)	79	136	215	0.7036 (0.1772)	0.7732 (0.1646)	<0.05
MBOT (Mucinous)	13	136	149	0.5032 (0.1590)	0.7731 (0.1643)	<0.05

^1^ SD, standard deviation.

**Table 2 ijms-22-04105-t002:** Accuracies of binary classification and prediction by machine learning.

Gene Set	Classification	Group	Sensitivity (SD ^1^)	Specificity (SD ^1^)	Accuracy (SD ^1^)	AUC ^2^
		All BOTs	0.9944 (0.01757)	1.0000 (0.0000)	0.9978 (0.0069)	0.9974
GO term	Binary	SBOT ^3^	0.9933 (0.02108)	1.0000 (0.0000)	0.9977 (0.0074)	0.9967
		MBOT ^4^	1.0000 (0.0000)	1.0000 (0.0000)	1.0000 (0.0000)	1.0000

^1^ SD, standard deviation; ^2^ AUC, area under curve; ^3^ SBOT, serous BOT; ^4^ MBOT, Mucinous BOT.

**Table 3 ijms-22-04105-t003:** The 50 most deregulated GO terms for all BOTs, SBOTs, and MBOTs ranked by CWI (cluster weight index).

Category	All BOTs			Serous BOTs			Mucinous BOTs		
Ranking	GO ID	GO Term	CWI	GO ID	GO Term	CWI	GO ID	GO Term	CWI
1	GO:0002682	Regulation of immune system process	0.02426	GO:0002682	Regulation of immune system process	0.0255	GO:0044281	Small molecule metabolic process	0.01957
2	GO:0001775	Cell activation	0.01783	GO:0005215	Transporter activity	0.02032	GO:0002682	Regulation of immune system process	0.01772
3	GO:0005215	Transporter activity	0.01737	GO:0001775	Cell activation	0.02015	GO:0051049	Regulation of transport	0.01434
4	GO:0006811	Ion transport	0.01678	GO:0006811	Ion transport	0.01765	GO:0006811	Ion transport	0.01187
5	GO:0044281	Small molecule metabolic process	0.01622	GO:0044281	Small molecule metabolic process	0.01653	GO:0045595	Regulation of cell differentiation	0.01145
6	GO:0051049	Regulation of transport	0.01345	GO:0051049	Regulation of transport	0.01324	GO:0006629	Lipid metabolic process	0.01135
7	GO:0002252	Immune effector process	0.01313	GO:0002252	Immune effector process	0.0131	GO:0001775	Cell activation	0.01114
8	GO:0001816	Cytokine production	0.01255	GO:0002520	Immune system development	0.01284	GO:0016070	RNA metabolic process	0.01101
9	GO:0002520	Immune system development	0.01219	GO:0001816	Cytokine production	0.01252	GO:0005215	Transporter activity	0.01008
10	GO:0045595	Regulation of cell differentiation	0.01147	GO:0045595	Regulation of cell differentiation	0.01129	GO:0001816	Cytokine production	0.00999
11	GO:0006629	Lipid metabolic process	0.01013	GO:0031399	Regulation of protein modification process	0.01021	GO:0022008	Neurogenesis	0.00984
12	GO:0040011	Locomotion	0.0094	GO:0006629	Lipid metabolic process	0.01018	GO:0040011	Locomotion	0.00954
13	GO:0031399	Regulation of protein modification process	0.00929	GO:0048585	Negative regulation of response to stimulus	0.00945	GO:0002520	Immune system development	0.00947
14	GO:0042592	Homeostatic process	0.00925	GO:0042592	Homeostatic process	0.00939	GO:0007267	Cell-cell signaling	0.00906
15	GO:0046903	Secretion	0.00922	GO:0022610	Biological adhesion	0.00926	GO:0042592	Homeostatic process	0.00884
16	GO:0007267	Cell-cell signaling	0.00903	GO:0055085	Transmembrane transport	0.00913	GO:0051276	Chromosome organization	0.0088
17	GO:0051240	Positive regulation of multicellular organismal process	0.00891	GO:0051240	Positive regulation of multicellular organismal process	0.00904	GO:0051240	Positive regulation of multicellular organismal process	0.00867
18	GO:0048585	Negative regulation of response to stimulus	0.00886	GO:0006915	Apoptotic process	0.00902	GO:0019219	Regulation of nucleobase-containing compound metabolic process	0.00864
19	GO:0022610	Biological adhesion	0.00873	GO:0046903	Secretion	0.00887	GO:0046903	Secretion	0.00858
20	GO:0016070	RNA metabolic process	0.00861	GO:0060429	Epithelium development	0.00875	GO:0060429	Epithelium development	0.00854
21	GO:0006915	Apoptotic process	0.00859	GO:0051174	Regulation of phosphorus metabolic process	0.00867	GO:0002252	Immune effector process	0.00826
22	GO:0051276	Chromosome organization	0.00839	GO:0051276	Chromosome organization	0.00866	GO:0033043	Regulation of organelle organization	0.00808
23	GO:0055085	Transmembrane transport	0.00818	GO:0007267	Cell-cell signaling	0.00846	GO:0006259	DNA metabolic process	0.00774
24	GO:0060429	Epithelium development	0.00807	GO:0007049	Cell cycle	0.0082	GO:0031399	Regulation of protein modification process	0.00767
25	GO:0007049	Cell cycle	0.00787	GO:0070727	Cellular macromolecule localization	0.008	GO:0000003	Reproduction	0.00742
26	GO:0051174	Regulation of phosphorus metabolic process	0.00779	GO:0016070	RNA metabolic process	0.00799	GO:0006915	Apoptotic process	0.00733
27	GO:0033043	Regulation of organelle organization	0.00771	GO:0040011	Locomotion	0.00776	GO:0007049	Cell cycle	0.00733
28	GO:0070727	Cellular macromolecule localization	0.00768	GO:0033043	Regulation of organelle organization	0.00766	GO:0048585	Negative regulation of response to stimulus	0.00725
29	GO:0046907	Intracellular transport	0.00753	GO:0046907	Intracellular transport	0.0076	GO:0009719	Response to endogenous stimulus	0.00719
30	GO:0019219	Regulation of nucleobase-containing compound metabolic process	0.00738	GO:0023056	Positive regulation of signaling	0.00728	GO:0019637	Organophosphate metabolic process	0.00714
31	GO:0023056	Positive regulation of signaling	0.00717	GO:0019219	Regulation of nucleobase-containing compound metabolic process	0.00705	GO:0070727	Cellular macromolecule localization	0.00689
32	GO:0022008	Neurogenesis	0.00698	GO:0006468	Protein phosphorylation	0.00701	GO:0051174	Regulation of phosphorus metabolic process	0.00673
33	GO:0005102	Signaling receptor binding	0.00669	GO:0051241	Negative regulation of multicellular organismal process	0.00661	GO:0030030	Cell projection organization	0.00646
34	GO:0006468	Protein phosphorylation	0.00668	GO:0006259	DNA metabolic process	0.00652	GO:0007010	Cytoskeleton organization	0.00642
35	GO:0006259	DNA metabolic process	0.00661	GO:0005102	Signaling receptor binding	0.00643	GO:0023056	Positive regulation of signaling	0.0064
36	GO:0051241	Negative regulation of multicellular organismal process	0.00635	GO:0098772	Molecular function regulator	0.00618	GO:0065003	Protein-containing complex assembly	0.00637
37	GO:0044419	Interspecies interaction between organisms	0.00635	GO:0080134	Regulation of response to stress	0.00617	GO:0007417	Central nervous system development	0.00628
38	GO:0098772	Molecular function regulator	0.00617	GO:0022008	Neurogenesis	0.00604	GO:0051241	Negative regulation of multicellular organismal process	0.00625
39	GO:0015849	Organic acid transport	0.00597	GO:0015849	Organic acid transport	0.00597	GO:0046907	Intracellular transport	0.00607
40	GO:0080134	Regulation of response to stress	0.00592	GO:0044419	Interspecies interaction between organisms	0.00593	GO:0018193	Peptidyl-amino acid modification	0.00591
41	GO:0009719	Response to endogenous stimulus	0.00578	GO:0023057	Negative regulation of signaling	0.00589	GO:0055085	Transmembrane transport	0.00572
42	GO:0002250	Adaptive immune response	0.00577	GO:0010941	Regulation of cell death	0.00588	GO:0080134	Regulation of response to stress	0.00564
43	GO:0000003	Reproduction	0.00571	GO:0042127	Regulation of cell proliferation	0.00583	GO:0005102	Signaling receptor binding	0.00558
44	GO:0018193	Peptidyl-amino acid modification	0.0057	GO:0009790	Embryo development	0.0058	GO:0051094	Positive regulation of developmental process	0.00555
45	GO:0023057	Negative regulation of signaling	0.00564	GO:0002250	Adaptive immune response	0.00579	GO:0022610	Biological adhesion	0.00552
46	GO:0030030	Cell projection organization	0.00557	GO:0019637	Organophosphate metabolic process	0.00557	GO:0044419	Interspecies interaction between organisms	0.00543
47	GO:0042127	Regulation of cell proliferation	0.00556	GO:0018193	Peptidyl-amino acid modification	0.00552	GO:0006468	Protein phosphorylation	0.0054
48	GO:0019637	Organophosphate metabolic process	0.00554	GO:0009628	Response to abiotic stimulus	0.00545	GO:0016788	Hydrolase activity, acting on ester bonds	0.00525
50	GO:0065003	Protein-containing complex assembly	0.00544	GO:0000003	Reproduction	0.00539	GO:0009790	Embryo development	0.0052

**Table 4 ijms-22-04105-t004:** The top 41 common deregulated GO terms among the three case groups (all BOTs, SBOTs, and MBOTs).

Order	GO ID	GO Term	B ^1^	S ^2^	M ^3^	Order	GO ID	GO Term	B ^1^	S ^2^	M ^3^
1	GO:0002682	Regulation of immune system process	1	1	2	21	GO:0060429	Epithelium development	24	20	20
2	GO:0044281	Small molecule metabolic process	5	5	1	22	GO:0006915	Apoptotic process	21	18	26
3	GO:0001775	Cell activation	2	3	7	23	GO:0007049	Cell cycle	25	24	27
4	GO:0006811	Ion transport	4	4	4	24	GO:0033043	Regulation of organelle organization	27	28	22
5	GO:0005215	Transporter activity	3	2	9	25	GO:0022610	Biological adhesion	19	15	45
6	GO:0051049	Regulation of transport	6	6	3	26	GO:0051174	Regulation of phosphorus metabolic process	26	21	32
7	GO:0045595	Regulation of cell differentiation	10	10	5	27	GO:0019219	Regulation of nucleobase-containing compound metabolic process	30	31	18
8	GO:0001816	Cytokine production	9	9	10	28	GO:0055085	Transmembrane transport	23	16	41
9	GO:0002520	Immune system development	8	8	13	29	GO:0022008	Neurogenesis	32	38	11
10	GO:0006629	Lipid metabolic process	11	12	6	30	GO:0070727	Cellular macromolecule localization	28	25	31
11	GO:0002252	Immune effector process	7	7	21	31	GO:0006259	DNA metabolic process	35	34	23
12	GO:0042592	Homeostatic process	14	14	15	32	GO:0023056	Positive regulation of signaling	31	30	35
13	GO:0031399	Regulation of protein modification process	13	11	24	33	GO:0046907	Intracellular transport	29	29	39
14	GO:0040011	Locomotion	12	27	12	34	GO:0051241	Negative regulation of multicellular organismal process	36	33	38
15	GO:0051240	Positive regulation of multicellular organismal process	17	17	17	35	GO:0005102	Signaling receptor binding	33	35	43
16	GO:0046903	Secretion	15	19	19	36	GO:0006468	Protein phosphorylation	34	32	47
17	GO:0007267	Cell-cell signaling	16	23	14	37	GO:0000003	Reproduction	43	49	25
18	GO:0016070	RNA metabolic process	20	26	8	38	GO:0080134	Regulation of response to stress	40	37	42
19	GO:0048585	Negative regulation of response to stimulus	18	13	28	39	GO:0009719	Response to endogenous stimulus	41	50	29
20	GO:0051276	Chromosome organization	22	22	16	40	GO:0044419	Interspecies interaction between organisms	37	40	46
						41	GO:0018193	Peptidyl-amino acid modification	44	47	40


^1^ B, Ranking in all BOTs; ^2^ S, Ranking in SBOTs; ^3^ M, Ranking in MBOTs.

**Table 5 ijms-22-04105-t005:** The top 20 most dysfunctional canonical pathways among all BOT, SBOT, and MBOT groups ranked by *p*-values.

Category	All BOTs		Serous BOT		Mucinous BOT	
Ranking	Geneset Name	*p*-Values	Geneset Name	*p*-Values	Geneset Name	*p*-Values
1	REACTOME galactose catabolism	1.9273 × 10^−34^	REACTOME galactose catabolism	8.1608 × 10^−32^	REACTOME glucuronidation	6.7592 × 10^−10^
2	BIOCARTA Th1Th2 pathway	3.6094 × 10^−33^	BAKER hematopoiesis STAT1 targets	5.6450 × 10^−29^	Reactome NOSTRIN mediated eNOS trafficking	1.9891 × 10^−9^
3	RODRIGUES thyroid carcinoma upregulation	3.6094 × 10^−33^	BIOCARTA Antigen dependent B Cell activation pathway	5.6450 × 10^−29^	CASTELLANO HRAS and NRAS targets downregulation	2.8989 × 10^−9^
4	ZHAN multiple myeloma downregulation	3.6094 × 10^−33^	BIOCARTA stem pathway	5.6450 × 10^−29^	REACTOME galactose catabolism	5.0812 × 10^−9^
5	BIOCARTA stem pathway	6.7888 × 10^−33^	KANG cisplatin resistance upregulation	5.6450 × 10^−29^	NADELLA PRKAR1A targets downregulation	5.3670 × 10^−9^
6	KANG cisplatin resistance upregulation	6.7888 × 10^−33^	REACTOME HATs acetylate histones	5.6450 × 10^−29^	REACTOME molybdenum cofactor biosynthesis	5.3670 × 10^−9^
7	TERAO AOX4 targets skin downregulation	6.7888 × 10^−33^	REACTOME interleukin 2 signaling	5.6450 × 10^−29^	REACTOME RUNX3 regulates BCL2 L11 (BIM) transcription	5.3670 × 10^−9^
8	WANG response to androgen upregulation	9.0067 × 10^−33^	REACTOME RUNX1 and FOXP3 control the development of regulatory T lymphocytes	5.6450 × 10^−29^	REACTOME synthesis of ketone bodies	5.3670 × 10^−9^
9	REACTOME HATs acetylate histones	9.6321 × 10^−33^	STAMBOLSKY targets of mutated TP53 upregulation	5.6450 × 10^−29^	GARGALOVIC response to oxidized phospholipids green module downregulation	1.5393 × 10^−8^
10	REACTOME interleukin 2 signaling	9.6321 × 10^−33^	TERAO AOX4 targets skin downregulation	5.6450 × 10^−29^	GAUSSMANN MLL-AF4 fusion proteins targets set B downregulation	1.5393 × 10^−8^
11	STAMBOLSKY targets of mutated TP53 upregulation	9.9179 × 10^−33^	WANG response to androgen upregulation	7.5300 × 10^−29^	KANG cisplatin resistance downregulation	1.5393 × 10^−8^
12	REACTOME RUNX1 and FOXP3 control the development of regulatory T lymphocytes	1.7040 × 10^−32^	HASEGAWA tumorigenesis by RET allele with the MEN2A mutation (C634R)	8.1498 × 10^−29^	REACTOME formyl peptide receptors bind formyl peptides and many other ligands	1.5393 × 10^−8^
13	JU aging TERC targets upregulation	4.1179 × 10^−32^	SCHWAB targets of BMYB polymorphic variants downregulation	8.1498 × 10^−29^	REACTOME mRNA editing	1.5393 × 10^−8^
14	BIOCARTA Antigen dependent B Cell activation pathway	5.6577 × 10^−32^	KREPPEL CD99 targets downregulation	1.1090 × 10^−28^	ABRAHAM amyloidosis plasma cells (ALPC) compared to multiple myeloma (MM) cells upregulation	1.6662 × 10^−8^
15	BIOCARTA 4-1BB (CD137) pathway	1.0825 × 10^−31^	NAGY STAGA complex components in human	1.5310 × 10^−28^	ACEVEDO liver cancer with H3K27 me3 upregulation	1.6662 × 10^−8^
16	LOPEZ mesothelioma survival downregulation	2.4323 × 10^−31^	WANG response to forskolin upregulation	2.1622 × 10^−28^	AMIT EGF response 20 min after stimulation of MCF10A cells	1.6662 × 10^−8^
17	BAKER hematopoiesis STAT1 targets	3.2802 × 10^−31^	PID JNK signaling in the CD4+ TCR pathway	3.1635 × 10^−28^	AMIT serum response 120 min after stimulation of MCF10A cells	1.6662 × 10^−8^
18	PID JNK signaling in the CD4+ TCR pathway	5.6082 × 10^−31^	REACTOME reversible hydration of carbon dioxide	5.8648 × 10^−28^	AMIT serum response 480 min after stimulation of MCF10A cells	1.6662 × 10^−8^
19	NAGY STAGA complex components in human	7.9222 × 10^−31^	BIOCARTA 4-1BB (CD137) pathway	7.8095 × 10^−28^	AMUNDSON response to sodium arsenite	1.6662 × 10^−8^
20	IKEDA miR-30 microRNA targets downregulation	1.0328 × 10^−30^	BIOCARTA Complement pathway	8.5975 × 10^−28^	AUNG gastric cancer	1.6662 × 10^−8^

## Data Availability

The datasesets of microarrays expression profiles are publically available in the National Center for Biotechnology Information (NCBI) Gene Expression Omnibus (GEO) repository (https://www.ncbi.nlm.nih.gov/geo/, accessed on 15 April 2021).

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
