# Peer review of "Dysregulated Immunological Functionome and Dysfunctional Metabolic Pathway Recognized for the Pathogenesis of Borderline Ovarian Tumors by Integrative Polygenic Analytics"

_ijms, 2021, doi:10.3390/ijms22084105_

Round 1
Reviewer 1 Report
The Authors of this paper have previously published on the topic of ovarian cancer pathophysiology with a particular attention to the gene expression profiles characterizing carcinogenesis process.
This study is an accurate gene-ontology analysis leading to draw an interesting picture of global functionomes and related pathogenic pathways relevant in cancer formation process.
Even though several intrinsic limitations characterize this study, as properly and clearly underlined by Authors in the Discussion section (i.e. the not homogeneous case group SBOTs and MBOTs), however, the data are adequately collected, carefully presented and sustained by accurate statistical analysis.
Author Response
Thanks for your kind comments and suggestions. We will keep it up diligently.Reviewer 2 Report
Summary
In the manuscript titled ‘Dysregulated immunological functionome and dysfunctional metabolic pathway recognized for the pathogenesis of border-line ovarian tumors by integrative polygenic analytics’ the authors have performed an analysis to determine markers that are unique for border-line ovarian tumors (BOT). They have looked into all BOTs, serous BOTs and mucinous BOTs. Through their integrative polygenic analysis, they were able to identify candidates that aid in the proper identification of the BOTs.
Strengths
The authors have explained the basis of the study and the methods in good detail. They have clearly explained the need for this study as well. In addition to their bioinformatics study, they tested two targets in diseased and control tissues by immunohistochemical analysis.
Weaknesses
There are not many weaknesses in the manuscript.
- It is not clear how the immunohistochemical samples were quantified. The authors mention measuring the intensity. Was this done using a software or visually. It would help to specify this aspect.
- There are many grammatical and linguistic errors in the manuscript. Additional proof reading will greatly enhance the quality of the manuscript.
Author Response
Point 1: It is not clear how the immunohistochemical samples were quantified. The authors mention measuring the intensity. Was this done using a software or visually. It would help to specify this aspect. Response 1: Thank you for constructive comment and suggestion. We had made clear explanations about how the samples were quantified via immunohistochemical analysis in materials and methods section 4.6. and all clinical samples were evaluated by the qualified and experienced pathologists. The immunohistochemistry (IHC) staining was assessed using a semi-quantitative method that multiplied the intensity (I) by the percentage of positive neoplastic cells (P) for the markers recruited in this study. The staining intensity was scored from 0 to 3 as following; 0 = no staining, 1 = weakly staining, 2 = moderately staining, and 3 = strongly expression. For the percentage of staining, it ranged from 0% to 100%. Hence, the formula for the IHC score (Q) = I x P; maximum = 300). The detailed results of all scores for GALT and IL among clinical samples are listed in Table S3 as attached. Table S3 All scores of GALT and IL of clinical samples Point 2: There are many grammatical and linguistic errors in the manuscript. Additional proof reading will greatly enhance the quality of the manuscript. Response 2: Thanks for your kind comment. This article has been revised and edited by American Journal Experts (www.aje.com) for English language editing. The attachment content has been attached with a certificate.
